# ON LAST-ITERATE CONVERGENCE OF DISTRIBUTED STOCHASTIC GRADIENT DESCENT ALGORITHM WITH MOMENTUM

## ABSTRACT

Distributed Stochastic Gradient optimization algorithms are studied extensively to address challenges in centralized approaches, such as data privacy, communication load, and computational efficiency, especially when dealing with large datasets. However, convergence theory research for these algorithms has been limited, particularly for distributed momentum-based SGD (mSGD) algorithms. Current theoretical work on distributed mSGD algorithms primarily focuses on establishing time-average convergence theory, whereas last-iterate convergence—considered a stronger and more practical definition than time-average convergence—has yet to be thoroughly explored. In this paper, we aim to establish the last-iterate convergence theory for a class of distributed mSGD algorithms with a decaying learning rate. First, we propose a general framework for distributed mSGD algorithms. Within this framework and under general conditions, we have proven the last-iterate convergence of the gradient of the loss function for a class of distributed mSGD algorithms. Furthermore, we have estimated the corresponding last-iterate convergence rate under supplementary conditions. Moreover, we theoretically prove that in the early stage, the adding of a momentum term can make the iterations converge more rapidly to a neighborhood of the stationary point. Some experiments are provided to illustrate the theoretical findings.

## 1 INTRODUCTION

As a typical stochastic gradient optimization algorithm, Stochastic Gradient Descend (SGD) Robbins & Monro (1951) has shown its prominent advantages, especially in the domain of deep learning. This is due to its effectiveness in handling large datasets and high-dimensional feature spaces effectively, such as regularized empirical risk minimization and training deep neural networks Graves et al. (2013); Nguyen et al. (2018); Hinton & Salakhutdinov (2006); Krizhevsky et al. (2012). Adding momentum to the SGD algorithm—an improvement known as momentum-based SGD (mSGD)—accelerates the convergence rate, as the accumulation of past gradient information helps reduce oscillations in complex optimization scenarios.Polyak (1964); Krizhevsky et al. (2012); Tang et al. (2018); Kim et al. (2014). Centralized stochastic gradient optimization algorithms, including the centralized mSGD and SGD, can be used to solve the optimization problems as follows:

$$\min_{x \in \mathbb{R}^N} \mathbb{E}_\xi \big( g(x, \xi) \big), \tag{1}$$

where $g(x, \xi)$ is defined as an unbiased estimate of the loss function $g(x)$ and $\xi$ is random noise induced by sampling or external disturbance. These centralized algorithms are designed for an architecture where a central server collects massive amounts of data from each edge devices, also known as work nodes, and performs gradient computation. However, this architecture may encounter several problems: 1) The data from edge devices may contain private information, making it infeasible to share raw data with the central server; 2) Transmitting large volumes of raw data, such as videos and images, can result in significant communication overloading. Furthermore, this architecture exhibits low computational efficiency, particularly for dealing with massive training datasets and complex deep neural network architectures.

To address these problems, many related distributed algorithms have been proposed. The idea of distributed algorithms is to establish cooperative training schemes among multiple worker nodes. In a distributed architecture, these algorithms compute gradients in parallel across each worker nodes and subsequently aggregate these gradients to update the model parameters. The distributed stochastic gradient optimization algorithms can be used to solve the following optimization problem in a distributed manner:

$$\min_{x \in \mathbb{R}^N} \mathbb{E}_\xi \left( g(x, \xi) \right), \quad g(x) = \frac{1}{m} \sum_{i=1}^m g_i(x), \tag{2}$$

where $m$ is the number of worker nodes and similar to a centralized manner in equation (1), $g(x, \xi)$ is defined as an unbiased estimate of the loss function $g(x)$, where $\xi$ represents random noise induced by sampling or external disturbance. Although distributed algorithms show its advantages in privacy preserving, reduced communication load and improved computational efficiency, the requirement for gradient communication between each worker node and either a central server, or its neighboring worker nodes remains. Consequently, these algorithms encounter communication delays, which may be influenced by various factors such as network congestion, bandwidth limitations, physical distance, and the performance of network hardware. This is especially true as increasingly heavy machine learning models, such as deep neural networks, are being utilized. Various communication-efficient techniques can be further integrated into distributed algorithms. Notably, periodic communication is a standout method that aims to reduce the frequency of communication rounds. The local-update SGD algorithmMcDonald et al. (2010), also known as Periodic Simple-Averaging SGD (PSASGD), allowed to perform local updates on each worker nodes and subsequently conduct periodic averaging of local model on each worker nodes. This approach reduce the total communication round significantly, thereby reducing communication delays. Unlike methods that perform a simple average of local models, the Elastic Averaging Stochastic Gradient Descent (EASGD) algorithmZhang et al. (2015) maintains an auxiliary variable that acts as an anchor during the update of local models on each worker nodes, preventing large deviations between local models during local updates. Another approach is to perform averaging of local models in a sparse-connected network topology, known as decentralized parallel SGD (D-PSGD) algorithmNedić et al. (2018). With D-PSGD algorithm, each node only needs to average its model with those of its neighbors, significantly reducing the communication complexity.

Rather than only focusing on the improvement of algorithms, it is equally important to understand their convergence properties. This understanding plays a key role in achieving effective and efficient training for a variety of machine learning models, including deep neural networks, Support Vector Machines (SVMs), logistic regression, and others. For PSASGD algorithm, the convergence has been studied for strongly convex objective functionsStich (2018) and for non-convex objectives with the assumption of uniformly bounded stochastic gradients at worker nodesYu et al. (2019c). Furthermore, the convergence of PSASGD for non-convex objectives has also been investigated without this boundedness assumption, by considering PSASGD as a special case of gradient sparsificationJiang & Agrawal (2018). For EASGD algorithm, the original paperZhang et al. (2015) provides a convergence analysis that is limited to the scenario with one local update for quadratic objective functions. The convergence of D-PSGD algorithm is studied for non-convex objective functions also in scenarios where workers are not permitted to perform more than one local updateLian et al. (2017b); Jiang et al. (2017); Zeng & Yin (2018). Recently, a general framework for distributed Stochastic Gradient Descent (SGD) algorithms has been proposed, named Cooperative SGDWang & Joshi (2021). This framework provides a unified convergence analysis for the class of Cooperative SGD algorithms, including the PSASGD, EASGD, and D-PSGD algorithms.

It is important to note that existing convergence analyses on distributed algorithms for solving problem equation 2 concentrate on distributed SGD without momentum. In practice, however, momentum SGD is more commonly used for training deep neural networks, as it often converges faster and generalizes betterKrizhevsky et al. (2012); Yan et al. (2018); Sutskever et al. (2013). From this perspective, there is a significant discrepancy between current practices—specifically, the preference for using momentum SGD over standard SGD in distributed training for deep neural networks—and the existing theoretical analyses, which primarily study the convergence rate and communication complexity of SGD without momentum. The only research on the convergence of distributed momentum-based stochastic gradient descent algorithms focuses on the time-average convergence theory for non-convex functionsYu et al. (2019b). There is no research on last-iterate

convergence, which is considered a stronger and more practical definition than time-average convergence.

In this paper, we aim to establish last-iterate convergence theory for a class of distributed mSGD algorithm, especially for Elastic Averaging SGD (EASGD) and Decentralized Parallel SGD (D-PSGD) algorithms with adding of momentum, with a decaying learning rate $\{\epsilon_n\}_{n\geq 0}$. The main contributions of this paper are summarized as follows:

- First, We develop a general framework for distributed mSGD algorithms that enables us to obtain a unified analysis. Within this framework and under general conditions, we prove the last-iterate almost-sure convergence and last-iterate mean-square convergence of the gradient of the loss function for a class of distributed mSGD algorithms which includes three popular distributed stochastic gradient descent algorithms in momentum form: Periodic Simple-Averaging SGD, Elastic Averaging SGD, and Decentralized Parallel SGD.

- Secondly, we estimate the corresponding last-iterate convergence rate under a mild supplementary condition.

- Finally, we prove that in the early stage, the adding of momentum term accelerate the rate at which iterations converge to a neighborhood of the stationary point. Additionally, we present a series of experiments designed to validate and illustrate our theoretical findings.

To our knowledge, these are the first results concerning the last-iterate convergence theory for the related algorithms, including momentum-based D-PSGD and momentum-based EASGD.

## 2 MAIN RESULTS

### 2.1 DEFINITIONS OF CONVERGENCE

For the problem equation 2, suppose the gradient of loss function $g_i(x)$ exists, which is denoted by $\nabla g(x)$. Then we say an iterate sequence $\{x_n\}$ ensures:

- *$\epsilon$-neighborhood time-average mean-square ($\epsilon$-TAMS) convergence* if given any scalar $\epsilon > 0$, such that after $n$ steps, it holds that $\frac{1}{n}\sum_{k=1}^{n}\mathbb{E}\left(\|\nabla g(x_k)\|^2\right) < \epsilon$;

- *Time-average mean-square (TAMS) convergence* if

$$\frac{1}{n}\sum_{k=1}^{n}\mathbb{E}\left(\|\nabla g(x_k)\|^2\right) = O(f(n)) \tag{3}$$

  with $f(n) \overset{n\to\infty}{\rightrightarrows} 0$;

- *Last-iterate mean-square (LIMS) convergence* if

$$\mathbb{E}\left(\|\nabla g(x_n)\|^2\right) = O(f(n)) \tag{4}$$

  with $f(n) \overset{n\to\infty}{\rightrightarrows} 0$;

- *Last-iterate almost-sure (LIAS) convergence* if

$$\|\nabla g(x_n)\| = O(f(n)) \tag{5}$$

  with $f(n) \overset{n\to\infty}{\rightrightarrows} 0$

We note that LIMS convergence can ensure TAMS convergence, but not vice versa. In addition, TAMS convergence can ensure $\epsilon$-TAMS convergence, but not vice versa.

### 2.2 GENERAL MOMENTUM-BASED ITERATION

First, we introduce two existing distributed SGD algorithms in the following.

**D-PSGD**. The decentralized SGD algorithm D-PSGD was studied in Jiang et al. (2021); Lin et al. (2018); Lian et al. (2017a); Wang & Joshi (2021). The idea is that each worker node performs local

updates and then conducts weighted model averaging with the models from neighboring worker nodes for every $k$ step, mathematically,

$$if \ n \quad \mod k = 0 :$$

$$x_{n+1}^{(i)} = \sum_{j=1}^{m} w_{ji} \big( x_n^{(j)} - \epsilon_n \nabla g_j(x_n^{(j)}, \xi_n^{(j)}) \big),$$

$$else :$$

$$x_{n+1}^{(i)} = x_n^{(i)} - \epsilon_n \nabla g_i(x_n^{(i)}, \xi_n^{(i)}),$$

where $x_n^{(i)}$ represents the model parameter of worker node $i$, $w_{ji}$ is the $(j,i)$-TH element of a mixing matrix $W$ indicating the influence of worker node $j$ in the weighted model averaging to worker node $i$. **PSASGD** corresponds to a special case of D-PSGD when the mixing matrix W has equal non-diagonal entries $w_{ji} = \frac{1}{m}$.

**EASGD**. In contrast to performing weighted model averaging of the local models in D-PSGD, the EASGD motivated by quadratic penalty method is to let each worker node keep its own local model first, and then use an update like elastic force to ensure that each worker node can coordinate its model with other worker nodes Zhang et al. (2015), mathematically,

$$if \ n \quad \mod k = 0 :$$

$$x_{n+1}^{(i)} = (1 - \beta)\big(x_n^{(i)} - \epsilon_n \nabla g_i(x_n^{(i)}, \xi_n^{(i)})\big) + \beta z_n$$

$$z_{n+1} = (1 - m\beta)z_n + m\beta\overline{x}_n,$$

$$eles :$$

$$x_{n+1}^{(i)} = x_n^{(i)} - \epsilon_n \nabla g_i(x_n^{(i)}, \xi_n^{(i)}),$$

$$z_{n+1} = z_n,$$

(6)

where $\overline{x}_n = \sum_{i=1}^{m} x_n^{(i)}/m$, and $\beta > 0$ is a parameter controlling the speed of consensus among all local models.

Authors in Wang & Joshi (2021) presented a general update rule of EASGD and D-PSGD as follows

$$X_{n+1} = W_n\big(X_n - \epsilon_n G(X_n, \xi_n)\big), \tag{7}$$

where for D-PSGD,

$$X_n = (x_n^{(1)}, x_n^{(2)}, ..., x_n^{(m)})^\top$$

$$G(X_n, \xi_n) = (\nabla g_1(x_n^1, \xi_n^{(1)}), \cdots, \nabla g_m(x_n^m, \xi_n^{(m)})))^\top$$

$$W_n = \begin{cases} (w_{ij})_{m \times m} & n \quad \mod k = 0 \\ \\ \mathbf{I}_m & n \quad \mod k \neq 0 \end{cases}.$$

and for EASGD,

$$X_n = (x_n^{(1)}, x_n^{(2)}, ..., x_n^{(m)}, z_n^{(1)}, z_n^{(2)}, ..., z_n^{(v)})^\top,$$

$$G(X_n, \xi_n) = (\nabla g_1(x_n^1, \xi_n^{(1)}), \cdots, \nabla g_m(x_n^m, \xi_n^{(m)}), \mathbf{0}, \cdots, \mathbf{0})^\top$$

$$W_n = \begin{cases} \begin{pmatrix} (1 - \beta)I & \beta\mathbf{1} \\ \beta\mathbf{1}^\top & 1 - m\beta \end{pmatrix} & n \quad \mod k = 0 \\ \\ \mathbf{I}_m & n \quad \mod k \neq 0 \end{cases}.$$

To accelerate the convergence rate of the EASGD and the D-PSGD by adding momentum, motivated by mSGD, one can modify equation equation 7 into the following iteration

$$v_n = \alpha v_{n-1} + \epsilon_n G(X_n, \xi_n),$$

$$X_{n+1} = W_n\big(X_n - v_n\big), \tag{8}$$

where $\alpha \in [0, 1)$ stands for the momentum coefficient and $\epsilon_n$ is the learning rate, and $v_0 := 0$. We note that the above iteration is reduced to the momentum-based D-PSGD in Yu et al. (2019b) when $W_n$ and $G(X)$ are set according to the D-PSGD. The algorithm equation equation 8 was also mentioned in Zhang et al. (2015); Yuan et al. (2021); Singh et al. (2021); Gao & Huang (2020); Balu et al. (2021); Yu et al. (2019b). Comparing with Algorithm 2 in Yu et al. (2019b), equation 8 does not have the procedure that each worker $i$ updates its local momentum term $v_n^{(i)}$ based on the ones of neighbors, i.e., $v_n \leftarrow W_n v_n$. In Zhang et al. (2015); Yuan et al. (2021); Singh et al. (2021); Gao & Huang (2020); Balu et al. (2021), equation 8 was also used. Meanwhile, there is no obvious difference in the techniques used to analyse the last-iterate convergence of the two different iterations. We denote $X = (x^{(1)}, x^{(2)}, ..., x^{(m)})$. For D-PSGD, let

$$G(X, \xi_n) = (\nabla g_1(x^{(1)}, \xi_n^{(1)}), \dots, \nabla g_m(x^{(m)}, \xi_n^{(m)})))^\top$$
$$G(X) = (\nabla g_1(x^{(1)}), \nabla g_2(x^{(2)}), ..., \nabla g_m(x^{(m)})))^\top,$$

and for EASGD, let

$$G(X, \xi_n) = (\nabla g_1(x^{(1)}, \xi_n^{(1)}), \cdots, \nabla g_m(x^{(m)}, \xi_n^{(m)}), \mathbf{0}, \cdots, \mathbf{0})^\top$$

$$G(x) = (\nabla g_1(x^{(1)}), \nabla g_2(x^{(2)}), ..., \nabla g_m(x^{(m)}), \mathbf{0}, \cdots, \mathbf{0})^\top.$$

In the following two sections, we will study the convergence of the general iteration equation 8.

## 2.3 Last-iterate Convergence

To proceed, the following assumptions are needed.

**Assumption 2.1.** $g(x) := \frac{1}{m} \sum_{i=1}^m g_i(x)$ *is a non-negative and continuously differentiable. In addition, the following conditions hold:*

1. *$G(X, \xi_n))$ is an unbiased estimate of $G(X)$, i.e., $\mathbb{E}_{\xi_n} G(X, \xi_n) = G(X)$;*

2. *The mixing matrix $W_n \in \mathbb{R}^{m \times m}$ is a symmetric doubly stochastic matrix with only one eigenvalue equal to one and the absolute values of the rest eigenvalues are less than one.*

3. *(Assumption 1 in Yu et al. (2019b)) There are two constants $L > 0$, $M > 0$, such that $\forall, X, Y \in \mathbb{R}^{m \times N}$, $\|G(X) - G(Y)\| \le L\|X - Y\|$ and $\|G(X)\| \le M$.*

4. *For any $i = 1, 2, ..., m$ and $\forall X \in \mathbb{R}^{m \times N}$, it holds that*

$$\sum_{i=1}^m \mathbb{E}_{\xi_n} \left\| \nabla g_i(x, \xi) - \nabla g_i(x) \right\|^2 \le \sigma_0^2.$$

*In addition, $\forall\, x \in \mathbb{R}^N$, it holds that*

$$\frac{1}{m} \sum_{i=1}^m \|\nabla g_i(x) - \nabla g(x)\|^2 \le \sigma_1^2.$$

The conditions in Assumption 2.1 are common in the study of distributed SGD or mSGD. We can find these conditions in the literature Yu et al. (2019b); Wang & Joshi (2021); Yu et al. (2019a); Jin et al. (2022b); Nguyen et al. (2018). In some works, the non-negative loss function condition may be replaced by a finite low bound condition, i.e., $g(x) > \hat{l}_0 > -\infty$. These two conditions are essentially equivalent, since one can construct a new loss function $\overline{g} = g - \hat{l}_0$ for the finite low bound condition, such that the new loss function is non-negative. Note that item 4 in Assumption 1 quantifies the variance of stochastic gradients at local worker, and $\sigma_1^2$ quantifies the deviations between the local objective function of each workers. The bounded variance assumption can be trivially generalized to the ABC growth condition, i.e., $\mathbb{E}_{\xi_n} \|\nabla g(x, \xi_n)\|^2 \le Ag(x) + B\|\nabla g(x)\|^2 + C$ ($A > 0$, $B > 0$, $C > 0$). For the sake of brevity in this proof, we did not consider this trivial generalization.

**Assumption 2.2.** *The momentum coefficient $\alpha \in [0, 1)$ and the sequence of learning rate $\epsilon_n$ satisfies the Robbins-Monro condition, i.e., it is positive, monotonically decreasing to zero, such that $\sum_{n=1}^{+\infty} \epsilon_n = +\infty$ and $\sum_{n=1}^{+\infty} \epsilon_n^2 < +\infty$.*

Assumption 2.2 means that a decreasing learning rate is required. Actually, for any stochastic optimal algorithm, due to the gradient noise $G(X_n, \xi_n) - G(X_n)$, decreasing learning rate is almost an essential condition to guarantee that last iterate can converge to stationary points Smith et al. (2017); Welling & Teh (2011); Khan et al. (2015); Gitman et al. (2019), i.e., $\nabla g(x_n) \to 0$ $a.s.$. This condition is common in the community of machine learning He et al. (2016); Yu et al. (2019b); Sutskever et al. (2013). In contrast, constant learning rate can just make the algorithm converge to a neighbor of stationary point (and not in the sense of last iteration), which indicates that the requirement $\sum_{n=1}^{+\infty} \epsilon_n^2 < +\infty$ is also reasonable and common in the literature, such as in Nguyen et al. (2018) for the convergence of SGD, and in Jin et al. (2022b) for the convergence of centralized mSGD.

Under the above assumptions, we attain the convergence of momentum-based distributed SGD as given in the following theorem.

**Theorem 2.1.** *Suppose $\{X_n\}$ is a sequence generated by equation equation 8. Under Assumptions 2.1–2.2, it holds that $\|\nabla g(\overline{x}_n)\| \to 0$ a.s. and $\mathbb{E}\|\nabla g(\overline{x}_n)\|^2 \to 0$, where $\overline{x}_n$ is defined as the average value of every worker node, i.e., $\overline{x}_n = 1/m \sum_{i=1}^m x_n^{(i)}$ .*

Our method is based on the work Jin et al. (2022b). Meanwhile, we have made some innovations to enhance this method, and enable its applicability to distributed problems. First, we have summarized the periodic communicated algorithm into a unified expression equation 7. We then eliminate the influence of the matrix $W_n$ in two steps by left-multiplying two different eigenvectors, reducing the problem to a centralized one. Second, our step 4 is more skilful and comprehensive compared with the approach in Jin et al. (2022b). In Jin et al. (2022b), authors attempted to prove the almost-sure convergence of the loss function sequence $\{g(\overline{x}_n)\}$ to imply the convergence of the gradient-norm sequence $\{\|\nabla g(\overline{x}_n)\|^2\}$ . However, this step is incomplete. For example, consider a saddle point $x$ where there exist many points connected to $x$ with non-zero gradient-norm and the same loss function value as $x$. Therefore, the convergence result $g(\overline{x}_n) \to g(x)$ $a.s.$ can only infer that $\overline{x}_n$ converges to this region, but this region has different gradient information, making that the convergence of gradient-norm cannot be inferred. Finally, we provide additional results on mean-square convergence, and we have revealed the intrinsic connection between these two types of convergence in Remark 1.

Theorem 2.1 accurately shows the last-iterate convergence of EASGD and D-PSGD, and our results imply the results with the time average form (described in Yu et al. (2019b); Yuan et al. (2021); Singh et al. (2021); Gao & Huang (2020); Balu et al. (2021)), i.e. $1/T \sum_{i=1}^{\top} \mathbb{E}\|\nabla g(\overline{x}_n)\|^2 \to 0$ .

## 2.4 LAST-ITERATE CONVERGENCE RATE

In general, if we need to quantitatively estimate the convergence rate of the last iterate, we usually need some extra assumptions. These assumptions are usually used to establish a quantitative relationship between $g$ and $\nabla g$. In the existing works, the strong-convex assumption is often required. For example, it was assumed in Yuan et al. (2021) that the loss function of each worker $g_i$ is strongly convex when studying the last-iterate convergence rate of deterministic distributed momentum-based GD. In addition, Nguyen et al. (2018) required that the loss function $g$ is strongly convex when studying the last-iterate convergence rate of SGD. In our paper, since Theorem 2.1 actually guarantees the asymptotic convergence, we just need a milder condition (compared with the above requirements) as follows:

**Assumption 2.3.** *The loss function $g(\theta)$ is a convex function and has a unique optimal point $\theta^*$.*

**Assumption 2.4.** *During the algorithm iteration process, stability is maintained, i.e., for any $n > 0$, there exists a constant $G < +\infty$ such that $\|u^\top X_n\| < G$ almost surely.*

Under these new assumptions, we can get the last-iterate convergence rate as follows:

**Theorem 2.2.** *Suppose $\{X_n\}$ is a sequence generated by equation equation 8. Under Assumptions 2.1– 2.4 with $\epsilon_n = \frac{\sqrt{m}}{\sqrt{n}}$. Then for any $T > 0$, there is*

$$\mathbb{E}(g(u^\top X_T) - g(\theta^*)) = \mathcal{O}\Big(\sqrt{m}\frac{\ln T}{\sqrt{T}}\Big) + \mathcal{O}\Big(\frac{1}{\sqrt{m}}\frac{\ln T}{\sqrt{T}}\Big).$$

It may be observed that including momentum does not significantly enhance the algorithm's convergence rate. This discrepancy is incongruous with experimental results that demonstrate momentum's

ability to expedite convergence. The reason for this inconsistency is that the convergence rate discussed here pertains to the asymptotic behavior as the number of epochs approaches infinity, whereas momentum primarily hastens the algorithm's progress during the initial stages. To formalize this effect, we present the following theorem.

**Theorem 2.3.** *Suppose $\{X_n\}$ is a sequence generated by equation equation 8. Under Assumption 2.1, given any non-increasing positive learning rate $\epsilon_n \geq \epsilon_{n+1}$ and bounded loss function, for any worker node $i$ $(i = 1, 2, ..., m)$, then for any $a_0 > 0$, any $V_0 \in \mathbb{R}^{mN}$ and any $\|\nabla g(\overline{x}_1)\|^2 > a_0$, there exists $s > 0$, such that*

$$P(\tau^{(a_0)} \geq n) = O\Big(e^{-\frac{s}{(1-\alpha)^2}\sum_{i=1}^n \epsilon_i}\Big),$$

*where $\tau^{(a_0)} = \min_{n>0}\{\|\nabla g_i(x_n)\|^2 < a_0\}$.*

**Remark 2.1.** *An intuitive understanding of why momentum can accelerate in the early stages (the gradients-norm is relatively large) can be explained as follows: when the gradients-norm is large, i.e., there exists a constant $d$ such that $\|\nabla g(\overline{x})\|^2 > d$, the random bias term $\mathbb{E}_{\xi_n}\|\nabla g(\overline{x}, \xi_n)\|^2$ can be bounded by the gradients-norm, i.e., $\mathbb{E}_{\xi_n}\|\nabla g(\overline{x}, \xi_n)\|^2 \leq \frac{\sigma_0^2}{d}\|\nabla g(\overline{x})\|^2$. This indicates that in the early stage, random noise approximately satisfies the strong growth condition. According to the results in Jin et al. (2022b), we can conclude that momentum can indeed accelerate the algorithm during this phase.*

Theorem 2.3 shows that a larger momentum term coefficient $\alpha$ can speed up the convergence in an early stage. In other words, given a scalar $\delta > 0$, a larger coefficient of the momentum term can make the first time instant of having $\|\nabla g(\overline{x}_n)\| \leq \delta$ become shorter. Denote the time instant by $\tau^{(a_0)}$, which is random in the stochastic setting. From Theorem 2.3, we see that a larger momentum term coefficient can have a larger probability such that $\|\nabla g(\overline{x}_{\tau^{(a_0)}})\| \leq \delta$ before a fixed time $n$. The reason why a larger momentum term coefficient generally does not guarantee a faster convergence rate over the whole time is that when time is sufficiently large, the upper bound of convergence rate is determined by the decreasing rate of learning rate $\epsilon_n$ (shown in Theorem 2.2).

## 3 EXPERIMENTS RESULTS

In this section, we consider a classification task where neural networks are trained using a distributed mSGD algorithm, to demonstrate the correctness of our theoretical findings.

**Implementation**. We employ the ResNet20 network using Keras. We initialize the weights using the Glorot uniform algorithm. The momentum coefficient takes on the values of 0, 0.5, and 0.9. We train the model using the categorical cross-entropy loss function. The learning rate begins at 0.1 and subsequently decays. We partition the dataset into three, ten, and twenty sub-datasets, with each sub-dataset communicating every 10 epochs with matrices $W$ defined as follows: $W = \frac{1}{3}\mathbf{1}_3^\top \mathbf{1}_3$. $W = \frac{1}{10}\mathbf{1}_{10}^\top \mathbf{1}_{10}$ and $W = \frac{1}{20}\mathbf{1}_{20}^\top \mathbf{1}_{20}$. The models are trained for up to 1000 epochs, which takes approximately two hours each time using a 3080 GPU. We do not incorporate dropouts in our training process.

**Dataset**. We use two distinct datasets: CIFAR-10 and CIFAR-100. Both datasets comprise 50,000 training images and 10,000 testing images. CIFAR-10 contains images across 10 classes, while CIFAR-100 spans 100 classes. These datasets are composed of color images depicting common objects, with each image measuring 32x32 pixels with 3 color channels. Each attribute of the data is normalized to $[0, 1]$.

**Results**. We conducted our experiments by using the distributed mSGD with three different momentum coefficients, namely, $\alpha = 0$ (corresponding to standard SGD), $\alpha = 0.5$, and $\alpha = 0.9$. The experimental results, as depicted in Figures 1 and 2, illustrate some key observations: The loss decreases to near zero across all three settings of the momentum coefficient, and the setting of $\alpha = 0.9$ results in the fastest convergence of the gradient of loss to a small neighborhood around zero, outperforming the other two settings. This empirical finding is in accordance with the theoretical analysis presented in Theorems 2.1 and 2.3."

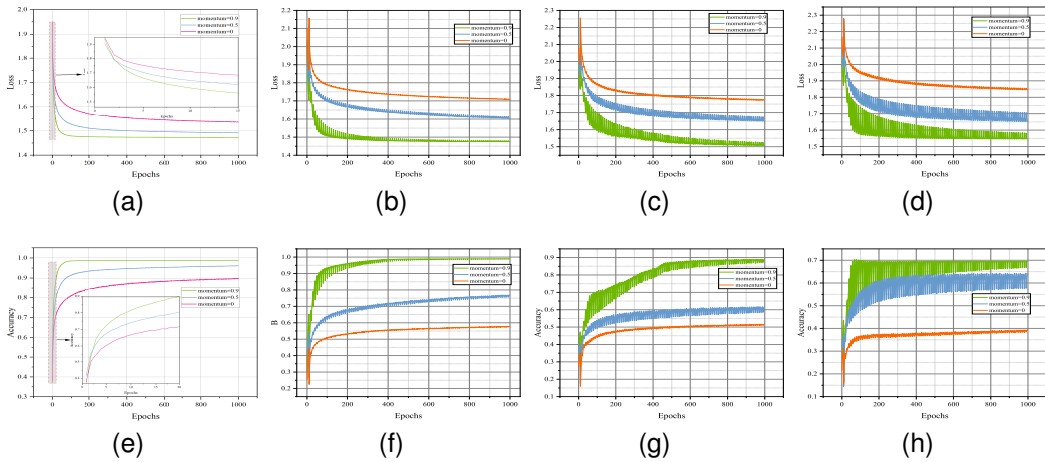

Figure 1: Training and prediction performance on CIFAR-10 with 1,3,10,20 sub-datasets (workers). (a)-(d): The training loss with 1, 3, 10, and 20 sub-datasets respectively. (e)-(h): The accuracy with 1, 3, 10, and 20 sub-datasets respectively.

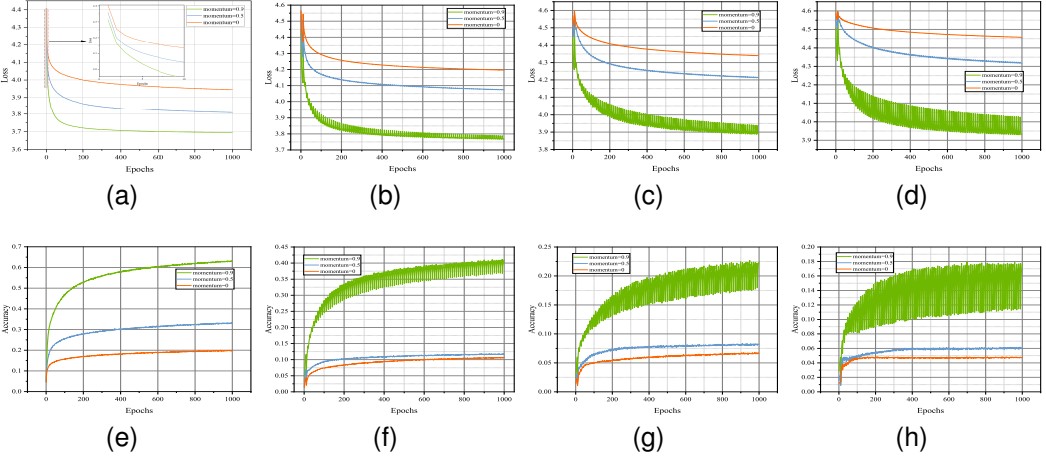

Figure 2: Training and prediction performance on CIFAR-100 with 1,3,10,20 sub-datasets (workers). (a)-(d): The training loss with 1, 3, 10, and 20 sub-datasets respectively. (e)-(h): The accuracy with 1, 3, 10, and 20 sub-datasets respectively.

## 4 CONCLUSION

This paper explores the last-iterate convergence for distributed mSGD algorithms. Our work addresses a critical gap in the current research by providing a thorough theoretical analysis of the last-iterate convergence properties of a class of distributed mSGD algorithms, with a decaying learning rate. Through the establishment of a general framework, we have proven the last-iterate almost-sure convergence and last-iterate mean-square convergence of the gradient of the loss function for a class of distributed mSGD algorithms, including momentum-based EASGD and momentum-based D-PSGD algorithms. Our findings indicate that adding a momentum term accelerates the convergence of iterations to a neighborhood of the stationary point in the early stages of the algorithm. Furthermore, under mild supplementary conditions, a larger momentum coefficient can lead to a higher convergence rate. These findings are important for understanding the performance of distributed mSGD algorithms in real-world applications. By showcasing the results of a classification tasks using ResNet20 network, which is optimized by the distributed mSGD algorithm, we find that

the experimental results are consistent with our theoretical findings. In conclusion, these theoretical results offer a substantial contribution to the field of distributed stochastic optimization, particularly in scenarios where communication efficiency and data privacy are of utmost importance.

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

# A  APPENDIX

## A.1  USEFUL LEMMAS

**Lemma A.1.** *(Lemma 1.2.3,Nesterov (2004)) Suppose $f(x) \in C^1$ $(x \in \mathbb{R}^N)$ with gradient satisfying the following Lipschitz condition*

$$\|\nabla f(x) - \nabla f(y)\| \le c\|x - y\|,$$

*then for any $x, y \in \mathbb{R}^N$, it holds that*

$$f(x) \le f(y) + \nabla f(y)^\top (x - y) + \frac{c}{2}\|x - y\|^2.$$

**Lemma A.2.** *(Lemma 10,Jin et al. (2022b)) Under the same conditions as Lemma A.1, for any $x_0 \in \mathbb{R}^N$, it holds that*

$$\left\|\nabla f(x_0)\right\|^2 \le 2c\big(f(x_0) - f^*\big),$$

*where $f^* = \inf_{x \in \mathbb{R}^N} f(x)$*

**Lemma A.3.** *(Lemma B.6 in Jin et al. (2022a)) If $0 < \mu < 1$ and $0 < \sigma < 1$ $(\sigma \ne \mu)$ are two constant, then exists $k_1 > 0$, $k_2 > 0$, for any positive sequence $\{\psi_n^{(i)}\}$, it holds that*

$$k_1 \sum_{i=1}^n \kappa^{n-i}\psi_i \le \sum_{k=1}^n \mu^{n-k}\sum_{i=1}^k \sigma^{k-i}\psi_i \le k_2 \sum_{i=1}^n \kappa^{n-i}\psi_i,$$

*where $\kappa = \max\{\mu, \sigma\}$ and $\omega_0 = \log_\kappa \min\{\mu, \sigma\}$.*

**Lemma A.4.** *If there exists a sequence of positive numbers $\{x_n\}_{n=1}^\infty$ such that $\sum_{n=1}^\infty x_n < \infty$, then for any $n > 0$, there exists a constant $k_n > 0$, uniform in $n$, such that for any $s$, it holds that $\sum_{k=s}^n x_k < k_n x_s$.*

**Lemma A.5.** *Wang et al. (2019) Suppose that $\{X_n\} \in \mathbb{R}^N$ is a $\mathcal{L}_2$ martingale difference sequence, and $(X_n, \mathcal{F}_n)$ is an adaptive process. Then it holds that $\sum_{k=0}^\infty X_k < +\infty$ a.s., if $\sum_{n=1}^\infty \mathbb{E}(\|X_n\|^2) < +\infty$ or $\sum_{n=1}^\infty \mathbb{E}\left(\|X_n\|^2\big|\mathcal{F}_{n-1}\right) < +\infty$.*

**Lemma A.6.** *(Lemma 6,Jin et al. (2022b)) Suppose that $\{X_n\} \in \mathbb{R}^N$ is a non-negative sequence of random variables, then it holds that $\sum_{n=0}^\infty X_n < +\infty$ a.s., if $\sum_{n=0}^\infty \mathbb{E}\left(X_n\right) < +\infty$.*

## A.2  PROOFS OF MAIN RESULTS

*Proof.* First, due to $0 < \alpha < 1$, we can always find a positive constant $\alpha_0$, making $\alpha_1 := \alpha^2 + \alpha_0 < 1$. Then based on the first equation of E.q. equation 8, we have

$$\|v_n\|^2 \le \alpha_1\|v_{n-1}\|^2 + \epsilon_n^2\Big(1 + \frac{1}{\alpha_0}\Big) \cdot \|G_{n,\xi_n}\|^2. \tag{9}$$

Then we take the mathematical expectation on the both side of E.q. equation 9, acquiring

$$\mathbb{E}\,\|v_n\|^2 \le \alpha_1 \cdot \mathbb{E}\,\|v_{n-1}\|^2 + \epsilon_n^2\Big(1 + \frac{1}{\alpha_0}\Big) \cdot \mathbb{E}\,\|G_{n,\xi_n} - G_n\|^2 + \epsilon_n^2\Big(1 + \frac{1}{\alpha_0}\Big) \cdot \mathbb{E}\,\|G_n\|^2$$

$$\le \alpha_1 \cdot \mathbb{E}\,\|v_{n-1}\|^2 + \epsilon_n^2\sigma_0^2\Big(1 + \frac{1}{\alpha_0}\Big) + \epsilon_n^2\Big(1 + \frac{1}{\alpha_0}\Big) \cdot \mathbb{E}\,\|G_n\|^2.$$

Iterating above inequity, we acquire

$$\mathbb{E} \|v_n\|^2 \le \alpha_1^n \cdot \|v_0\|^2 + \epsilon_n^2 \Big(1 + \frac{1}{\alpha_0}\Big) \cdot \sum_{s=1}^{n} (\sigma_0^2 + \mathbb{E} \|G_s\|^2) \cdot \alpha_1^{n-s} \tag{10}$$

Next, we iterate the second equation of E.q. equation 8 to attain

$$\begin{aligned}
X_{n+1} &= W_n X_n - W_n v_n \\
&= W_n \big(W_{n-1} X_{n-1} - W_{n-1} v_{n-1}\big) - W_n v_n \\
&= W_n W_{n-1}(W_{n-2} X_{n-2} - W_{n-2} v_{n-2}) \\
&\quad - W_n W_{n-1} v_{n-1} - W_n v_n \\
&= ... = \Big(\prod_{s=1}^{n} W_s\Big) \cdot X_1 - \sum_{t=1}^{n} \Big(\Big(\prod_{s=t}^{n} W_s\Big) \cdot v_t\Big).
\end{aligned}$$

We left multiply both sides of the above equation by the vector $e_i^\top := (-1/m, -1/m, ..., 1 - 1/m, ..., -1/m, ..., -1/m)$ ( the $i$-th entry is $1 - 1/m$, and others are $1/m$) to obtain

$$e_i^\top X_{n+1} = e_i^\top \Big(\prod_{s=1}^{n} W_s\Big) \cdot X_1 - \sum_{t=1}^{n} \Big(e_i^\top \Big(\prod_{s=t}^{n} W_s\Big) \cdot v_t\Big). \tag{11}$$

When $s \mod k = 0$, according to assumption2.1 2), we can find an orthogonal matrix $Q$ such that $Q^\top W_s Q = \mathrm{diag}\{1, \lambda_2, \lambda_3, \dots, \lambda_m\}$, where $\lambda_0 := \max_{2 \le j \le m}\{|\lambda_j|\} < 1$. When $n \mod k \ne 0$, we always have $W_s = \mathbf{I}_m = QQ^\top$. We assign $W_{t,n} := \prod_{s=t}^{n} W_s$. Then we can get that

$$\begin{aligned}
W_{t,n} &= \prod_{s\in[t,n],\ s \mod k=0} W_s \\
&= \prod_{s\in[t,n],\ s \mod k=0} (Q \cdot \mathrm{diag}\{1, \lambda_2, \lambda_3, \dots, \lambda_m\} \cdot Q^\top) \\
&= Q \cdot \mathrm{diag}\Big\{1, \prod_{s\in[t,n],\ s \mod k=0} \lambda_2, \\
&\qquad \prod_{s\in[t,n],\ s \mod k=0} \lambda_3, \dots, \prod_{s\in[t,n],\ s \mod k=0} \lambda_m\Big\} \cdot Q^\top.
\end{aligned} \tag{12}$$

We can conclude that $\forall\, j \in [2, m]$, there is

$$\Big| \prod_{s\in[t,n],\ s \mod k=0} \lambda_j \Big| \le \lambda_0^{c(t,n)}, \tag{13}$$

where $c(t, n)$ represents the total number of integers divisible by $k$ between $t$ and $n$. It is easy to prove that

$$\Big\lfloor \frac{n-t}{k} \Big\rfloor \le c(t,n) \le \Big\lfloor \frac{n-t}{k} \Big\rfloor + 1.$$

Then based on E.q. equation 12, we can derive the following expression:

$$\|e_i^\top X_{n+1}\|^2 \le 2\|e_i^\top W_{1,n}\|^2 \cdot \|X_1\|^2 + 2\Big( \sum_{t=1}^{n} \|e_i^\top W_{t,n}\| \cdot \|v_t\| \Big)^2. \tag{14}$$

For any $t$ and $n$, since the matrix $W_{t,n}$ is a real symmetric matrix, its eigenspaces are orthogonal to each other. We know that $(1, 1, ..., 1)^\top$ is obviously an eigenvector corresponding to the eigenvalue 1, and according to Assumption 2.1 2), we know that the dimension of the eigenspace corresponding to the eigenvalue 1 can only be 1. Therefore, the eigenspace corresponding to the eigenvalue 1 is completely spanned by the vector $(1, 1, ..., 1)^\top$. On the other hand, since $e_i^\top (1, 1, ..., 1)^\top =$

0, we know that $e_i$ must belong to the direct sum of the eigenspaces of $W_{t,n}$ other than the one corresponding to eigenvalue 1. Hence, there exists an orthogonal decomposition

$$e_i = r_2 e_{2,i} + r_3 e_{3,i} + ... + r_m e_{m,i},$$

where each $e_{s,i}$ ($2 \leq s \leq m$) is a unit vector and at the same time an eigenvector of the matrix $W_{t,n}$ corresponding to an eigenspace not associated with the eigenvalue 1. Therefore, we can obtain

$$\|e_i^\top W_{t,n}\| = \left\| \sum_{s=2}^m r_s e_{s,i}^\top W_{t,n} \right\| = \left( \sum_{s=2}^m |r_s^2| \right)^{\frac{1}{2}} \cdot \lambda_0^{c(t,n)} \tag{15}$$
$$= \|e_i\| \lambda_0^{c(t,n)}.$$

Substitute above inequity into E.q. equation 14, getting

$$\|e_i^\top X_{n+1}\|^2 \leq 2\|e_i\|^2 \cdot \|X_1\|^2 \cdot \lambda_0^{2c(1,n)} + 2\|e_i\|^2 \left( \sum_{t=1}^n \lambda_0^{c(t,n)} \cdot \|v_t\| \right)^2$$

$$\leq 2\|e_i\|^2 \cdot \|X_1\|^2 \cdot \lambda_0^{c(1,n)} + 2\|e_i\|^2 \lambda(n,k) \sum_{t=1}^n \lambda_0^{c(t,n)} \cdot \|v_t\|^2,$$

where $\lambda(n,k) = \sum_{t=1}^n \lambda_0^{c(t,n)}$. We take the mathematical expectation, resulting

$$\mathbb{E}\|e_i^\top X_{n+1}\|^2 \leq 2\|e_i\|^2 \cdot \|X_1\|^2 \cdot \lambda_0^{c(1,n)} + 2\|e_i\|^2 \cdot \lambda(n,k) \sum_{t=1}^n \lambda_0^{c(t,n)} \cdot \mathbb{E}\|v_t\|^2. \tag{16}$$

We substitute equation 10 into equation 16, getting

$$\mathbb{E}\|e_i^\top X_{n+1}\|^2 \leq 2\|e_i\|^2 \cdot \|X_1\|^2 \cdot \lambda_0^{c(1,n)} + 2\|e_i\|^2 \cdot \lambda(n,k)$$

$$\sum_{t=1}^n \lambda_0^{c(t,n)} \cdot \left( \alpha_1^t \|v_0\|^2 + \epsilon_t^2 \left( 1 + \frac{1}{\alpha_0} \right) \sum_{s=1}^t (\sigma_0^2 + \mathbb{E}\|G_s\|^2) \cdot \alpha_1^{t-s} \right)$$

$$= 2\|e_i\|^2 \cdot \|X_1\|^2 \cdot \lambda_0^{c(1,n)} + 2\|e_i\|^2 \lambda(n,k) \sum_{t=1}^n \lambda_0^{c(t,n)} \cdot \alpha_1^t \cdot \|v_0\|^2$$

$$+ 2\|e_i\|^2 \lambda(n,k) \sum_{t=1}^n \lambda_0^{c(t,n)} \cdot \epsilon_t^2 \left( 1 + \frac{1}{\alpha_0} \right) \sum_{s=1}^t (\sigma_0^2 + M^2) \alpha_1^{t-s}.$$

In the above inequality, by substituting the estimate for $c(t,n)$ from equation 13 and simplifying, we can obtain

$$\mathbb{E}\|e_i^\top X_{n+1}\|^2 = O\left( \sum_{t=1}^n \max\{\lambda_0^{\frac{1}{k}}, \alpha_1\}^{n-t} \cdot \epsilon_t^2 \right) \to 0. \tag{17}$$

We recall E.q. equation 8 as follows

$$v_n = \alpha v_{n-1} + \epsilon_n G(X_n, \xi_n),$$
$$X_{n+1} = W_n(X_n - v_n).$$

Then we multiply $u^\top = (1/m, 1/m, ..., 1/m)$ on the both sides of the above equalities to obtain

$$u^\top v_n = \alpha u^\top v_{n-1} + \epsilon_n u^\top G(X_n, \xi_n),$$
$$u^\top X_{n+1} = u^\top W_n(X_n - v_n).$$

Since $W_n$ is a doubly stochastic matrix, $u^\top W_n = u^\top$, Furthermore, it holds that

$$u^\top v_n = \alpha u^\top v_{n-1} + \epsilon_n u^\top G(X_n, \xi_n),$$
$$u^\top X_{n+1} = u^\top X_n - u^\top v_n.$$

Denote $\mathbf{I}_m = [1, 1, \cdots, 1]_{1 \times m}$ and $\otimes$ is Kronecker Product. We derive $g(u^\top X_{n+1}) - g(u^\top X_n)$ to obtain

$$g(u^\top X_{n+1}) - g(u^\top X_n) = -(u^\top G(\mathbf{I}_m \otimes u^\top X_n))^\top (u^\top v_n) + (u^\top G(\mathbf{I}_m \otimes u^\top X_n) - u^\top G(\mathbf{I}_m \otimes u^\top X_{\zeta_n}))^\top (u^\top v_n)$$
$$\leq -(u^\top G(\mathbf{I}_m \otimes u^\top X_n))^\top (u^\top v_n) + L \|u^\top v_n\|^2,$$
(18)

where $u^\top X_{\zeta_n}$ is a value between $u^\top X_n$ and $u^\top X_{n+1}$. Next we focus on the term $(u^\top G(\mathbf{I}_m \otimes u^\top X_n))^\top (u^\top v_n)$. We derive that

$$(u^\top G(\mathbf{I}_m \otimes u^\top X_n))^\top (u^\top v_n) = (u^\top G(\mathbf{I}_m \otimes u^\top X_n))^\top (\alpha u^\top v_{n-1} + \epsilon_n u^\top G(X_n, \xi_n))$$
$$= \alpha(u^\top G(\mathbf{I}_m \otimes u^\top X_n))^\top u^\top v_{n-1} + \epsilon_n (u^\top G(\mathbf{I}_m \otimes u^\top X_n))^\top u^\top G(X_n, \xi_n))$$
$$\geq \alpha u^\top G(\mathbf{I}_m \otimes u^\top X_{n-1})^\top (u^\top v_{n-1}) - L\|u^\top v_{n-1}\|^2 + \epsilon_n (\mathbf{I}_m \otimes u^\top G(X_n))^\top u^\top G(X_n, \xi_n)).$$
(19)

It follows from E.q. equation 19 that

$$(u^\top G(\mathbf{I}_m \otimes u^\top X_n))^\top (u^\top v_n) \geq -L \sum_{s=0}^{n-1} \alpha^{n-s-1} \|u^\top v_s\|^2 + \sum_{s=1}^{n} \alpha^{n-s} \epsilon_s (\mathbf{I}_m \otimes u^\top G(X_s))^\top u^\top G(X_s, \xi_s)).$$
(20)

Substituting E.q. equation 20 into E.q. equation 18 leads to

$$g(u^\top X_{n+1}) - g(u^\top X_n) \leq L \sum_{s=1}^{n} \alpha^{n-s} \|u^\top v_s\|^2 - \sum_{s=1}^{n} \alpha^{n-s} \epsilon_s (\mathbf{I}_m \otimes u^\top G(X_s))^\top u^\top G(X_s, \xi_s)) + L\|u^\top v_0\|^2 \cdot \alpha^n.$$
(21)

Then we consider the term $(u^\top G(\mathbf{I}_m \otimes u^\top X_s))^\top u^\top G(X_s, \xi_s))$ to have

$$- (u^\top G(\mathbf{I}_m \otimes u^\top X_s))^\top u^\top G(X_s, \xi_s))$$
$$= -u^\top G(\mathbf{I}_m \otimes u^\top X_s)^\top (u^\top G(X_s, \xi_s) - u^\top G(X_s)) - \|(u^\top G(\mathbf{I}_m \otimes u^\top X_s))^\top\|^2$$
$$+ u^\top G(\mathbf{I}_m \otimes u^\top X_s)^\top (u^\top G(\mathbf{I}_m \otimes u^\top X_s) - u^\top G(X_s))$$
$$\leq -\frac{1}{2} \|u^\top G(\mathbf{I}_m \otimes u^\top X_s)\|^2 + 2L \sum_{i=1}^{m} \|x_s^{(i)} - u^\top X_s\|^2 - u^\top G(\mathbf{I}_m \otimes u^\top X_s)^\top (u^\top G(X_s, \xi_s) - u^\top G(X_s)).$$
(22)

Denote $\beta_s := 2L \sum_{i=1}^{m} \|x_s^{(i)} - u^\top X_s\|^2$, then substituting E.q. equation 22 into E.q. equation 21 yields

$$g(u^\top X_{n+1}) - g(u^\top X_n) \leq L \sum_{s=1}^{n} \alpha^{n-s} \|u^\top v_s\|^2 - \frac{1}{2} \sum_{s=1}^{n} \alpha^{n-s} \epsilon_s \|u^\top G(\mathbf{I}_m \otimes u^\top X_s)\|^2 + \sum_{s=1}^{n} \alpha^{n-s} \epsilon_s \beta_s$$
$$- \sum_{s=1}^{n} \alpha^{n-s} \epsilon_s u^\top G(\mathbf{I}_m \otimes u^\top X_s)^\top \cdot (u^\top G(X_s, \xi_s) - u^\top G(X_s)) + L\|u^\top v_0\|^2 \cdot \alpha^n.$$
(23)

On the other hand, we have

$$\|u^\top v_n\|^2 = \|\alpha u^\top v_{n-1} + \epsilon_n u^\top G(X_n, \xi_n)\|^2$$
$$= \alpha^2 \|u^\top v_{n-1}\|^2 + 2\alpha \epsilon_n (u^\top v_{n-1})^\top u^\top G(X_n, \xi_n) + \epsilon_n^2 \|u^\top G(X_n, \xi_n)\|^2$$
(24)
$$= \alpha^2 \|u^\top v_{n-1}\|^2 + 2\alpha \epsilon_n (u^\top v_{n-1})^\top u^\top G(X_n) + \epsilon_n^2 \|u^\top G(X_n, \xi_n)\|^2 + \gamma_n,$$

where $\gamma_n = 2\alpha \epsilon_n v_{n-1}^\top u^\top (G(X_n, \xi_n) - G(X_n))$. Then we calculate $2\epsilon_n(E.q.equation\ 18 - E.q.equation\ 19) + E.q.equation\ 24$ to obtain

$$2\epsilon_{n+1}g(u^\top X_{n+1}) - 2\epsilon_n g(u^\top X_n)$$

$$\leq \alpha^2\|u^\top v_{n-1}\|^2 - \|u^\top v_n\|^2 + 2\epsilon_n L\|u^\top v_n\|^2 + \hat{\gamma}_n + \epsilon_n^2\|u^\top G(X_n, \xi_n)\|^2 - 2\epsilon_n^2(u^\top G(\mathbf{I}_m \otimes u^\top X_n))^\top u^\top G(X_n, \xi_n)$$

$$\leq \alpha^2\|u^\top v_{n-1}\|^2 - \|u^\top v_n\|^2 + 2\epsilon_n L(\|u^\top v_n\|^2 + \|u^\top v_{n-1}\|^2) + \epsilon_n^2\|u^\top G(X_n, \xi_n)\|^2$$

$$+ \hat{\gamma}_n - \epsilon_n^2\|u^\top G(\mathbf{I}_m \otimes u^\top X_n)\|^2 + 2\epsilon_n^2\beta_n,$$

$$\tag{25}$$

where

$$\hat{\gamma}_n := \gamma_n + 2\epsilon_n^2(u^\top G(\mathbf{I}_m \otimes u^\top X_n))^\top \left(u^\top G(X_n, \xi_n) - u^\top G(X_n)\right).$$

We make the mathematical expectation of E.q. equation 23 to obtain

$$\mathbb{E}\left(g(u^\top X_{n+1})\right) - \mathbb{E}\left(g(u^\top X_n)\right) \leq \hat{L}\sum_{s=1}^n \alpha^{n-s}\,\mathbb{E}\left\|u^\top v_s\right\|^2 - \frac{1}{2}\sum_{s=1}^n \alpha^{n-s}\epsilon_s\,\mathbb{E}\left\|u^\top G(\mathbf{I}_m \otimes u^\top X_s)\right\|^2$$

$$+ L\|u^\top v_0\| \cdot \alpha^n + \sum_{s=1}^n \alpha^{n-s}\epsilon_s\beta_s.$$

Making a summation of the above inequality leads to

$$\mathbb{E}\left(g(u^\top X_{n+1})\right) - \mathbb{E}\left(g(u^\top X_1)\right) \leq \frac{L}{1-\alpha}\sum_{s=1}^n \mathbb{E}\left\|u^\top v_s\right\|^2 - \frac{1}{2}\sum_{s=1}^n \epsilon_s\,\mathbb{E}\left\|u^\top G(\mathbf{I}_m \otimes u^\top X_s)\right\|^2 + \frac{L\|u^\top v_0\|^2}{1-\alpha} + \hat{\beta}_n,$$

$$\tag{26}$$

where $\hat{\beta}_n = \sum_{t=1}^n \sum_{s=1}^\top \alpha^{t-s}\epsilon_s\beta_s$. We perform the same operations on E.q. equation 25 to obtain

$$2\epsilon_{n+1}\,\mathbb{E}\left(g(u^\top X_{n+1})\right) - 2\epsilon_1\,\mathbb{E}\left(g(u^\top X_1)\right) \leq -\sum_{s=1}^n(1-\alpha^2)\,\mathbb{E}\left\|u^\top v_s\right\|^2 + \sum_{s=1}^n \epsilon_s^2\,\mathbb{E}\left\|u^\top G(X_s, \xi_s)\right\|^2 + 2\sum_{s=1}^n \epsilon_s^2\beta_s.$$

$$\tag{27}$$

For the term $\sum_{s=1}^n \epsilon_s^2\,\mathbb{E}\|u^\top G(X_s, \xi_s)\|^2$, we have

$$\sum_{s=1}^n \epsilon_s^2\,\mathbb{E}\left\|u^\top G(X_s, \xi_s)\right\|^2 \leq 2\sum_{s=1}^n \epsilon_s^2\|u^\top G(X_s, \xi_s) - u^\top G(X_s)\|^2 + 2\sum_{s=1}^n \epsilon_s^2\,\mathbb{E}\left\|u^\top G(X_s)\right\|^2$$

$$\leq 2\sum_{s=1}^n \epsilon_s^2\|u^\top G(X_s, \xi_s) - u^\top G(X_s)\|^2 + 4\sum_{s=1}^n \epsilon_s^2\,\mathbb{E}\left\|u^\top G(X_s) - u^\top G(\mathbf{I}_m \otimes u^\top X_s)\right\|^2$$

$$+ 4\sum_{s=1}^n \epsilon_s^2\,\mathbb{E}\left\|u^\top G(\mathbf{I}_m \otimes u^\top X_s)\right\|^2.$$

From Assumption 2.1 Item (4), we know that

$$2\sum_{s=1}^n \epsilon_s^2\|u^\top G(X_s, \xi_s) - u^\top G(X_s)\|^2 + 4\sum_{s=1}^n \epsilon_s^2\,\mathbb{E}\left\|u^\top G(X_s) - u^\top G(\mathbf{I}_m \otimes u^\top X_s)\right\|^2 \leq 2(\sigma_0^2 + 2L\sigma_1)\sum_{s=1}^n \epsilon_s^2,$$

which means

$$\sum_{s=1}^n \epsilon_s^2\,\mathbb{E}\left\|u^\top G(X_s, \xi_s)\right\|^2 \leq 2(\sigma_0^2 + 2L\sigma_1)\sum_{s=1}^n \epsilon_s^2 + 4\sum_{s=1}^n \epsilon_s^2\,\mathbb{E}\left\|u^\top G(\mathbf{I}_m \otimes u^\top X_s)\right\|^2.$$

Substitute above inequity into E.q. equation 27, getting

$$2\epsilon_{n+1} \mathbb{E}\left(g(u^\top X_{n+1})\right) - 2\epsilon_1 \mathbb{E}\left(g(u^\top X_1)\right) \leq -\sum_{s=1}^{n}(1-\alpha^2)\mathbb{E}\|u^\top v_s\|^2 + 2(\sigma_0^2 + 2\sigma_1)\sum_{s=1}^{n}\epsilon_s^2$$

$$+ 4\sum_{s=1}^{n}\epsilon_s^2 \mathbb{E}\|u^\top G(\mathbf{I}_m \otimes u^\top X_s)\|^2 + 2\sum_{s=1}^{n}\epsilon_s^2\beta_s.$$

(28)

We calculate $\frac{1-\alpha}{L}E.q.equation\ 26 + \frac{1}{1-\alpha^2}E.q.equation\ 28$, from Assumption 2.1 4) and E.q. equation 17 ($\sum_{s=1}^{n}\epsilon_s^2\beta_s \to 0$, $\hat{\beta}_n \to 0$), we can get

$$\sum_{s=1}^{+\infty}\epsilon_s \mathbb{E}\|\nabla g(\overline{x}_n)\|^2 < +\infty, \quad \sum_{s=1}^{+\infty}\epsilon_s\|\nabla g(\overline{x}_n)\|^2 < +\infty\ a.s.,$$

where the second inequity is because Lemma A.6. Then by using the condition $\sum_{n=1}^{+\infty}\epsilon_n = +\infty$, we can immediately acquire

$$\liminf_{n\to+\infty}\mathbb{E}\|\nabla g(\overline{x}_n)\|^2 = 0, \quad \liminf_{n\to+\infty}\|\nabla g(\overline{x}_n)\|^2 = 0\ a.s.\ .$$

Our goal below is to prove

$$\limsup_{n\to+\infty}\mathbb{E}\|\nabla g(\overline{x}_n)\|^2 = 0, \quad \limsup_{n\to+\infty}\|\nabla g(\overline{x}_n)\|^2 = 0\ a.s.\ .$$

We first prove $\limsup_{n\to+\infty}\|\nabla g(\overline{x}_n)\|^2 = 0\ a.s.\ $. We use proof by contradiction. We assume that for a certain trajectory $\{\|\nabla g(\overline{x}_n)\|^2\}_{n=1}^{+\infty}$, apart from 0, there exists another accumulation point $\hat{u} > 0$. Then, for a certain open interval $(o, e) \subset (0, \hat{u})$, the sequence $\{\|\nabla g(\overline{x}_n)\|^2\}_{n=1}^{+\infty}$ must cross this interval infinitely many times. We denote all the intervals that go upwards as $\{(\|\nabla g(\overline{x}_{l_n})\|^2, \|\nabla g(\overline{x}_{r_n})\|^2)\}_{n=1}^{+\infty}$. We have

$$\sum_{n=1}^{+\infty}\sum_{i=l_n}^{r_n}\epsilon_i < \frac{1}{o}\sum_{n=1}^{+\infty}\sum_{i=l_n}^{r_n}\epsilon_i\|\nabla g(\overline{x}_i)\|^2 < +\infty.$$

(29)

On the other hand, due to $\|\nabla g(\overline{x}_{r_n})\|^2 > e$ and $\|\nabla g(\overline{x}_{l_n})\|^2 < e$, we know there is a $\tilde{p}_0 > 0$, such that $\|\theta_{r_n} - \theta_{l_n}\| > \tilde{p}_0$. Then we get

$$\tilde{p}_0 < \|\theta_{r_n} - \theta_{l_n}\| = \zeta_n + k_0\sum_{i=l_n}^{r_n}\epsilon_i,$$

where $\zeta_n \to 0$. We get

$$\liminf_{n\to+\infty}\sum_{i=l_n}^{r_n}\epsilon_i > \frac{\tilde{p}_0}{2k_0} > 0,$$

which conclude

$$\sum_{n=1}^{+\infty}\sum_{i=l_n}^{r_n}\epsilon_i = +\infty.$$

(30)

Now we have a contradiction between E.q. equation 30 and E.q. equation 29, which implies that our assumption is false. Therefore, we obtain $\limsup_{n\to+\infty}\|\nabla g(\overline{x}_n)\|^2 = 0\ a.s.$, that is $\lim_{n\to+\infty}\|\nabla g(\overline{x}_n)\|^2 = 0\ a.s.\ $. Using the same technique, we can obtain convergence in the mean square sense, i.e., $\lim_{n\to+\infty}\mathbb{E}\|\nabla g(\theta_n)\|^2 = 0$ from the inequity $\sum_{s=1}^{+\infty}\epsilon_s \mathbb{E}\|\nabla g(\overline{x}_n)\|^2 < +\infty$. □

### A.3 PROOF OF THEOREM 2.2

*Proof.* We define $z_n = \frac{u^\top X_n - \alpha u^\top X_{n-1}}{1-\alpha}$. We can obtain $\forall\, \theta_0 \in \mathbb{R}^d$ which satisfies $\|z_n - \theta_0\| \leq \tau$, the following recursive inequality:

$$\|z_{n+1} - \theta_0\|^2 = \|z_n - \theta_0 + z_{n+1} - z_n\|^2 = \|z_n - \theta_0\|^2 + 2(z_n - \theta_0)^\top(z_{n+1} - z_n) + \|z_{n+1} - z_n\|^2. \tag{31}$$

Due to the definition of $z_{n+1} - z_n$, we have

$$z_{n+1} - z_n = \frac{u^\top(X_{n+1} - X_n) - \alpha u^\top(X_n - X_{n-1})}{1-\alpha}$$
$$= \frac{-u^\top v_n + \alpha u^\top v_{n-1}}{1-\alpha}$$
$$= -\frac{\epsilon_n u^\top G(X_n, \xi_n)}{1-\alpha}.$$

Substitute above equation into Eq. equation 31, and take the mathematical expectation, noting $\mathbb{E}(G(X_n, \xi_n)) = \mathbb{E}(G(X_n))$, getting

$$\mathbb{E}\|z_{n+1} - \theta_0\|^2 = \mathbb{E}\|z_n - \theta_0\|^2 - \frac{2\epsilon_n}{1-\alpha} \cdot \mathbb{E}\left((z_n - \theta_0)^\top u^\top G(X_n)\right) + \frac{\epsilon_n^2}{(1-\alpha)^2}\mathbb{E}\|u^\top G(X_n, \xi_n)\|^2. \tag{32}$$

For $u^\top G(X_n)$, dur to Eq. equation 17, we get

$$u^\top G(X_n) = \nabla g(u^\top X_n) + (u^\top G(X_n) - \nabla g(u^\top X_n))$$
$$= \nabla g(z_n) + \frac{\alpha}{1-\alpha}(\nabla g(u^\top X_n) - \nabla g(z_n)) + (u^\top G(X_n) - \nabla g(u^\top X_n)) + .$$

Then we get

$$-\frac{2\epsilon_n}{1-\alpha} \cdot \mathbb{E}\left((z_n - \theta_0)^\top u^\top G(X_n)\right) \leq -\frac{2\epsilon_n}{1-\alpha} \cdot \mathbb{E}\left((z_n - \theta_0)^\top \nabla g(z_n)\right) + \mathcal{O}(\epsilon_n^2).$$

Substitute above inequity into Eq. equation 31, acquiring

$$\mathbb{E}\|z_{n+1} - \theta_0\|^2 = \mathbb{E}\|z_n - \theta_0\|^2 - \frac{2\epsilon_n}{1-\alpha} \cdot \mathbb{E}\left((z_n - \theta_0)^\top \nabla g(z_n)\right) + \mathcal{O}(\epsilon_n^2). \tag{33}$$

For any term $k$ in the first $T$ iterations $1, 2, \ldots, T$, we set $\theta_0$ in Eq. equation 31 to $z_{T-k}$, obtaining $\exists\, l > 0,\ l_0 > 0$ such that

$$\sum_{t=T-k}^{T} \mathbb{E}((z_t - z_{T-k})^\top \nabla g(z_n)) \leq \frac{l}{\sqrt{m}}\left(\sqrt{T} - \sqrt{T-k}\right) + l_0\sqrt{m}\sum_{t=T-k}^{T}\frac{1}{\sqrt{t}}.$$

By convexity, we can lower bound $(z_t - z_{T-k})^\top \nabla g(z_t)$ bt $g(z_t) - g(z_{T-k})$. Also, it is easy to get that

$$\sum_{t=T-k}^{T}\frac{1}{\sqrt{t}} \leq 2(\sqrt{T} - \sqrt{T-k-1}).$$

Then we get

$$\mathbb{E}\left(\sum_{t=T-k}^{T}(g(z_t) - g(z_{T-k}))\right) \leq \left(\frac{l}{\sqrt{m}} + l_0\sqrt{m}\right)\left(\sqrt{T} - \sqrt{T-k-1}\right) \leq \left(\frac{l}{\sqrt{m}} + l_0\sqrt{m}\right)\frac{k+1}{\sqrt{T}}.$$

Then we define $S_k = \frac{1}{k+1}\sum_{t=T-k}^{T}\mathbb{E}(g(z_t))$ be the expected average value of the last $K+1$ iterates. The bound above implies that

$$-\mathbb{E}(g(z_{T-k})) \leq -\mathbb{E}(S_k) + \frac{l\sqrt{m} + \frac{l_0}{\sqrt{m}}}{\sqrt{T}}.$$

By the definition of $S_k$ and the inequity above, we have

$$k\,\mathbb{E}(S_{k-1}) = (k+1)\,\mathbb{E}(S_k) - \mathbb{E}(g(z_{T-k})) \leq (k+1)\,\mathbb{E}(S_k) - \mathbb{E}(S_k) + \frac{l\sqrt{m} + \frac{l_0}{\sqrt{m}}}{\sqrt{T}},$$

and dividing by $k$, implies

$$\mathbb{E}(S_{k-1}) \leq \mathbb{E}(S_k) + \frac{l\sqrt{m} + \frac{l_0}{\sqrt{m}}}{k\sqrt{T}}.$$

Using the inequity repeatedly and by summing over $k = 1, ..., T-1$, we have

$$\mathbb{E}(g(z_T)) = \mathbb{E}(S_0) \leq \mathbb{E}(S_{T-1}) + \frac{l\sqrt{m} + \frac{l_0}{\sqrt{m}}}{\sqrt{T}} \sum_{k=1}^{T-1} \frac{1}{k}.$$

Using Eq. equation 33 with $k = T - 1$ and $\theta_0 = \theta^*$, we can get

$$\mathbb{E}(S_{T-1}) - g(\theta^*) \leq \frac{l\sqrt{m} + \frac{l_0}{\sqrt{m}}}{\sqrt{T}}.$$

Finally, we get

$$\mathbb{E}(g(u^\top X_T) - g(\theta^*)) = \mathcal{O}\left(\left(l\sqrt{m} + \frac{l_0}{\sqrt{m}}\right) \frac{\ln T}{\sqrt{T}}\right).$$

$\square$

### A.4 PROOF OF THEOREM 2.3

We define another event

$$B_n = \{\|\nabla g(\overline{x}_1)\|^2 > a_0, \|\nabla g(\overline{x}_2)\|^2 > a_0 \cdots, \|\nabla g(\overline{x}_n)\|^2 > a_0\},$$

and its characteristic function as $I_n^{(a_0)}$. Then through Assumption 2.1 and $\epsilon_n \geq \epsilon_{n+1}$ we get that

$$I_{n+1}^{(a_0)} g(\overline{x}_{n+1}) - I_n^{(a_0)} g(\overline{x}_n) = -\frac{1}{2} \sum_{k=i}^{n} \alpha^{n-k} \epsilon_k I_k^{(a_0)} \|\nabla g(\overline{x}_k)\|^2 + \frac{\hat{\mu}_0 \sigma_0^2}{2} \sum_{k=i}^{n} \alpha^{n-k} I_k^{(a_0)} O(\epsilon_k^2) + \overline{k} \alpha^n + \zeta_n$$

$$+ \hat{L} \sum_{k=1}^{n} \alpha^{n-k} I_k^{(a_0)} \epsilon_k \beta_k,$$

where $\overline{k}$, $\hat{L}$ and $\hat{\mu}_0$ are three constants which can not affect the result. Notice that

$$I_k^{(a_0)} O(\epsilon_k^2) \leq \frac{1}{a_0} I_k^{(a_0)} \|\nabla g(\overline{x}_k)\|^2 O(\epsilon_k^2)$$

$$= I_k^{(a_0)} \|\nabla g(\overline{x}_k)\|^2 O(\epsilon_k^2).$$

Then we get

$$I_{n+1}^{(a_0)} g(\overline{x}_{n+1}) - I_n^{(a_0)} g(\overline{x}_n) = -\frac{1}{2} \sum_{k=i}^{n} \alpha^{n-k} \left(\epsilon_k - O(\epsilon_k^2)\right) \mathbb{E}\left(I_k^{(a_0)} \|\nabla g(\overline{x}_k)\|^2\right) + \zeta_n.$$

Due to $\mathbb{E}(\zeta_n) = 0$, we make the mathematical expectation to obtain

$$\mathbb{E}\left(I_{n+1}^{(a_0)} g(\overline{x}_{n+1})\right) - \mathbb{E}\left(I_n^{(a_0)} g(\overline{x}_n)\right) = -\frac{1}{2} \sum_{k=i}^{n} \alpha^{n-k} \left(\epsilon_k - O(\epsilon_k^2)\right) I_k^{(a_0)} \|\nabla g(\overline{x}_k)\|^2.$$

We denote

$$\hat{F}_n^{(a_0)} = \sum_{i=1}^{n} \left(\frac{1}{2-\alpha}\right)^{n-i} \mathbb{E}\left(I_n^{(a_0)} g(\theta_n)\right).$$

For convenience, we let

$$\hat{G}_n^{(a_0)} = \frac{2}{(1-\alpha)^2} \sum_{i=1}^{n} \left(\frac{1}{2-\alpha}\right)^{n-i} (\epsilon_i - O(\epsilon_i^2))$$
$$\cdot \mathbb{E}\left(I_i^{(a_0)} \|\nabla g(\overline{x}_n)\|^2\right).$$

Then we get

$$\hat{F}_{n+1}^{(a_0)} - \hat{F}_n^{(a_0)} \leq -\hat{G}_n^{(a_0)},$$

so there is

$$\hat{F}_{n+1}^{(a_0)} \leq \hat{F}_n^{(a_0)} \left(1 - \frac{\hat{G}_n^{(a_0)}}{\hat{F}_n^{(a_0)}}\right) \leq \hat{F}_1^{(a_0)} \prod_{i=1}^{n} \left(1 - \frac{\hat{G}_i^{(a_0)}}{\hat{F}_i^{(a_0)}}\right) \leq \hat{q}_0 \prod_{i=1}^{n} \left(1 - \frac{\hat{G}_i^{(a_0)}}{\hat{F}_i^{(a_0)}}\right),$$

where $\hat{q}_0$ is a constant. We focus on $\frac{\hat{G}_i^{(a_0)}}{\hat{F}_i^{(a_0)}}$. Using $O'stolz\ theorem$ yields

$$\liminf_{i \to +\infty} \frac{\hat{G}_i^{(a_0)}}{\epsilon_i \hat{F}_i^{(a_0)}} = \liminf_{i \to +\infty} \frac{2}{(1-\alpha)^2} \frac{\sum_{t=1}^{i}(\frac{1}{2-\alpha})^{i-t} \mathbb{E}(I_t^{(a_0)} \|\nabla g(\overline{x}_t)\|^2)}{\sum_{t=1}^{i}(\frac{1}{2-\alpha})^{i-t} \mathbb{E}(I_t^{(a_0)} g(\overline{x}_t))}$$
$$\geq \liminf_{i \to +\infty} \frac{2}{(1-\alpha)^2} \frac{\mathbb{E}(I_i^{(a_0)} \|\nabla g(\overline{x}_i)\|^2)}{\mathbb{E}(I_t^{(a_0)} g(\overline{x}_i))}.$$

In the setting of this theorem, the loss function is bounded. We let $g(x) < \hat{T}$. Then there is

$$\liminf_{i \to +\infty} \frac{2}{(1-\alpha)^2} \frac{\mathbb{E}(I_i^{(a_0)} \|\nabla g(\overline{x}_i)\|^2)}{\mathbb{E}(I_t^{(a_0)} g(\overline{x}_i))} \geq \liminf_{i \to +\infty} \frac{2}{(1-\alpha)^2} \frac{a}{\hat{T}}.$$

Then it holds that

$$\mathbb{E}(I_n^{(a_0)}) \leq \hat{F}_{n+1}^{(a_0)} = O\left(e^{-\frac{s}{(1-\alpha)^2} \sum_{i=1}^{n} \epsilon_n}\right),$$

where $s = 2a/\hat{T}$.

