# OpenReview forum: "On last-iterate convergence of distributed Stochastic Gradient Descent algorithm with momentum"
_ICLR.cc/2025/Conference — ICLR 2025 Conference Withdrawn Submission_

### Official Review · Reviewer_ETGm · 2024-10-21

**Soundness:** 2
**Presentation:** 1
**Contribution:** 2
**Rating:** 3
**Confidence:** 3

**Summary:**

This paper gives last-iterate convergence results for SGD with momentum in the distributed setting. The two main algorithms studied are D-PSGD and EASGD, but basically any algorithm which fits in the form
- update the momentum at each step,
- apply the momentum at each step, possibly with (partial/local) averaging,
as described by Eq (8) in the paper, is analyzed. Results for convergence in gradient norm are first given (without rates), then a function value result (with rates) is given. Finally, a last result on the hitting time for level sets of gradient norm is given, to demonstrate that the probability that this hitting time is above a given threshold reduces when the momentum increases. Experimental results compare the performance of distributed SGD when varying the momentum parameter and the averaging frequency.

**Strengths:**

- Extends last-iterate convergence results
- Seems to fix issues in existing results (paragraph after Theorem 2.1).

**Weaknesses:**

- Many typos (highlighted a few) and loosely defined quantities, which make the paper hard to read.
- Results are derived under (very) strong assumptions, assuming both Lipschitzness and boundedness of the gradients for instance, but also that iterates remain bounded almost surely (Assumption 2.4). This should be proven, as it depends on the algorithms, not assumed.
- Results are very coarse, in the sense that the impact of the distributed aspect is hardly investigated. Theorems give the same results for both D-PSGD and EASGD. This is because theorems are asymptotic, so that if one combines diminishing step-sizes with strong regularity assumptions then these distributed algorithms asymptotically behave as mini-batch SGD (the algorithms moves more and more slowly due to the step-size, but averaging is still performed at the same frequency).
- Results (both theory and experiments) make it look like the higher the momentum the better the results, with no upper bound on this.
- Experiments do not really have added value since this is a very classical setup, and the point of the paper is about last-iterate convergence, not comparing different values for momentum.


In the end, there is some value in these results, but rewriting is necessary to clarify the paper, higlight dependencies problem parameters (matrix W, momentum), and better justify the assumptions (is 2.4 technical or necessary? If technical it should be removed, otherwise it should be explained why one needs it).  Non-asymptotic guarantees would also be highly appreciated.

Typos/reformulation needed:

Incorrect citations: no space and ~\citet{} instead of \citep{}
Typos:
213: equation equation 7
261: low bound condition
281: equation equation
many other equation equation

Definitions of convergence: not very rigorous in the way they are defined. For instance I understand that $\epsilon$-TAMS reads: for any given scalar $\epsilon > 0$, there exists $n>0$ (which depends on $\epsilon$) such that... but it's not how it's written.

Assumption 2.1: I guess "function" is missing?

What is u in Assumption 2.4 and Theorem 2.2?

$m \geq 1$, so the second term is useless in Theorem 2.2.

Figures and their legends are very small and hard to read.

**Questions:**

1 ) In the experiments, the best momentum parameter identified is .9, but I guess at some point increasing the momentum degrades performances. What happens with a momentum of 1? Similarly, the way Theorem 2.3 is formulated is sketchy in that regard: it looks like taking $\alpha =1$ just gives "instant convergence", which has to be wrong. I believe this is due to the fact of expressing a hitting time probability with a $O()$ notation (hiding for instance $a_0$), but this should be clarified.

2) How does matrix W impact the results?

3) Is it possible to lift Assumption 2.4?

---

### Official Review · Reviewer_zwEu · 2024-11-03

**Soundness:** 3
**Presentation:** 3
**Contribution:** 1
**Rating:** 3
**Confidence:** 3

**Summary:**

This paper develops last-iterate convergence theory for distributed momentum-based SGD (mSGD) algorithms with a decaying learning rate, addressing limitations of time-average convergence in existing work. The paper established asymptotic convergence of the gradient norm under bounded gradient and Lipschitz continuous gradient assumptions. It also establishes convergence rate results under more strengthened assumptions.

**Strengths:**

I agree with the paper’s point that it is a limit of many existing works that the convergence and convergence rates results are often only for average gradient norm or similar. So the motivation and potential of the work seems solid. The paper is also well written and the mathematical results seem correct.

**Weaknesses:**

a) To me, the results are not significant enough for justifying a publication in a top ML conference. Firstly, the main results Theorem 2.1 do not provide convergence rate results, it is just asymptotic convergence that is ensured. The convergence rate results in Theorem 2.2 are almost useless, see my comment c). Given that convergence and convergence rate is already established for these algorithms, just showing that the last iterate converges feels marginal.

b) Assumption 2.1(b), which states that the gradient is bounded, is quite strong; however, it is not the reason for my low score. But it means, e.g., that quadratic and strongly convex functions are not covered by the analysis.

c) The convergence rate results in Theorem 2.2 are under very strange assumptions. Firstly, Assumption 2.4 cannot be checked before running the algorithm, so it seems quite useless. Secondly, by Assumption 2.3 the function should be convex and have a unique optimizer. But it cannot be strongly convex, since by Assumption 2.1 the gradient should be bounded. I am not sure what functions satisfy these assumptions.

d) I don’t see the point with the experiment. The algorithms have already been studied and shown to converge, and numerically investigated, it is unclear what is the message? Also there any baselines and it is unclear if the assumptions of the theorems are satisfied for the considered setups.

**Questions:**

In Theorem 2.3, I don’t understand the definition of \tau^{(a_0)}. I am guessing there is a typo, it should be the set of indexes where the condition holds? Or, alternatively, argmin.

It is difficult to read the results in figures 1 and 2. Firstly, the legends are so small, it is impossible to read without significantly zooming in. Secondly, the figures appear to be excessively large; when I zoom in, my computer struggles to handle the display smoothly, nearly causing it to freeze.

In section 2.4, given that the function is convex with a unique optimal solution, why consider only convergence of the gradient? It should be possible to translate these bounds to ||\theta^k-\theta^|| or objective function value f(\theta^k)-f*. Also, why use \theta for objective value in Section 2.4, when x was used everywhere else?

---

### Official Review · Reviewer_9eGj · 2024-11-06

**Soundness:** 3
**Presentation:** 3
**Contribution:** 3
**Rating:** 5
**Confidence:** 4

**Summary:**

This paper provides a general framework for decentralized SGD with local momentum steps. The authors provide last-iterate convergence analyses and analysis on the effect of momentum coefficient. Experiments demonstrate the effects of momentum coefficients.

**Strengths:**

1. This paper studies an important problem, that is the last-iterate convergence, in contrast to randomized or (min of so far) type of convergence, last-iterate can save computation and is more relevant to practice.
2. The paper provides detailed convergence analyses and characterizations of the effects of the momentum conefficients.

**Weaknesses:**

1. The literature review part misses some relevant works. In reviewing decentralized SGD, the authors reviewed existing works of decentralized SGD with single local update (line 94-95), and it seems that the authors are missing relevant works that allow multiple local updates such as [1] , and [2] with additional gradient tracking.
2. To establish last iterate convergence, this paper assumes bounded stochastic gradient, which is kind of strong, given that even the milder condition of gradient similarity, is not assumed in methods like SCAFFOLD [3], and gradient tracking [2]. What challenges might arise in extending the current analysis under relaxed conditions?
3. Could the authors clarify if Lemma A.2 requires convexity? If not, how does it hold for non-convex functions like $\sin x$?.
4. The writing of this paper sometimes is confusing to me that some terms coming up without definition. See my questions.
5. Could the authors provide a more detailed explanation of how Theorem 2.3 specifically relates to acceleration in the early stages, perhaps with an illustrative example or intuition, since it seems to it holds for all $n$?
6. The bounded averaged iterate assumption in Assumption 2.4 also seems to me very strong, the boundedness of the iterates should be the result of analysis, not a prior assumption. Is this assumption common in related works and how can one relax this assumption possibly?
7. The framework is proposed to subsume three special cases, so I would suggest that the authors include experiments for the other two cases or explain why chose to focus on only one case in the experiments. This would help readers better understand the generality and applicability of the proposed framework.

References
1. Li, X., Yang, W., Wang, S., & Zhang, Z. (2019). Communication efficient decentralized training with multiple local updates. stat, 1050, 21.
2. Ge, S., & Chang, T. H. (2023, December). Gradient Tracking with Multiple Local SGD for Decentralized Non-Convex Learning. In 2023 62nd IEEE Conference on Decision and Control (CDC) (pp. 133-138). IEEE.
3. Karimireddy, S. P., Kale, S., Mohri, M., Reddi, S., Stich, S., & Suresh, A. T. (2020, November). Scaffold: Stochastic controlled averaging for federated learning. In International conference on machine learning (pp. 5132-5143). PMLR.
4. Beck, A. (2017). First-order methods in optimization. Society for Industrial and Applied Mathematics.

**Questions:**

1. Can you explicitly add the additional condition to your contribution bullet point 2?
2. I think the the second part of assumption 2.1.4 is implied by the boundedness assumptions in 2.1.3, assuming that the authors mean Frobenius norm for the matrix represented stochastic gradients.
3. In assumption 2.4, what's the $u$ here? Do you mean for any $u \in \mathbb{R}^d$? Can I take $u =  (1/m)\mathbf{1}$, then as long as the global average is bounded during the optimization process, then we can obtain a last iterate rate for the global average?
4. In Theorem 2.3, do you mean $x_n^{(i)}$ or $\bar{x}_n$, not clear to me. What is $V_0$?
5. In Theorem 2.3, by bounded, do you mean lower bounded or upper bounded?
6. Can the authors briefly discuss what tricks used in this paper enable the analysis of last iterate convergence?

---

### Official Review · Reviewer_kwt1 · 2024-11-06

**Soundness:** 2
**Presentation:** 1
**Contribution:** 1
**Rating:** 3
**Confidence:** 4

**Summary:**

The paper addresses distributed stochastic optimization and introduces a framework for distributed momentum SGD (mSGD) by integrating momentum steps into existing distributed SGD algorithms. The authors provide theoretical results, establishing last-iterate convergence for convex objectives. Additionally, numerical experiments are presented to validate the theoretical findings.

**Strengths:**

- The paper establishes last-iterate convergence for distributed stochastic optimization, which is a stronger guarantee than the time-average mean-square convergence.
- The paper demonstrates that incorporating a momentum term can accelerate the algorithm's rate of convergence during the early stages of optimization.

**Weaknesses:**

- The proposed framework lacks sufficient novelty. It essentially extends existing algorithms by adding momentum steps, which does not introduce a significant new contribution.
- The last-iterate convergence results and the observation that the momentum term can accelerate the algorithm in the early stages are not novel and can be traced back to the work of Jin et al. (2022b). It appears that the extension is merely a straightforward modification without the introduction of any new techniques or insights..
- Limited theoretical scope: The analysis is restricted to convex functions, which significantly limits the generalizability of the results.
- Assumption 2.4 appears overly restrictive and may not hold in many practical scenarios.
- The experiments presented in the paper are overly simplistic and lack sufficient depth. The experiment only tests one kind of algorithms and its performance under different $\alpha$.
- The clarity of the presentation needs significant improvement. There are numerous typographical errors and unclear formulations throughout the paper

[1] On the convergence of mSGD and AdaGrad for stochastic optimization. In International Conference on Learning Representations, 2022b.

**Questions:**

- It appears that the result presented in Theorem 2.2 does not lead to a linear speedup. Could the authors provide further clarification or additional analysis to explain why this is the case
- Refer to the weakness part.

**Details Of Ethics Concerns:**

N.A.

---

### Note · Authors · 2024-11-13

**Comment:**

I am writing to request the withdrawal of our paper due to the presence of numerous typographical errors that have significantly impacted the clarity and accuracy of the content.

We understand the importance of maintaining the integrity and reliability of the scholarly record, and we believe that the best course of action is to retract the article to prevent the dissemination of incorrect information. We apologize for any inconvenience this may cause and appreciate your understanding in this matter.

Furthermore, we will take steps to thoroughly review and correct the manuscript before considering resubmission to your esteemed journal or another appropriate publication venue.

Thank you for your attention to this matter.

**Withdrawal Confirmation:**

I have read and agree with the venue's withdrawal policy on behalf of myself and my co-authors.